# Repurposing Nitazoxanide for Potential Treatment of Rare Disease Lymphangioleiomyomatosis

**DOI:** 10.3390/biom14101236

**Published:** 2024-09-30

**Authors:** Stella Bähr, Ryan W. Rue, Carly J. Smith, Jillian F. Evans, Hubert Köster, Vera P. Krymskaya, Dirk Pleimes

**Affiliations:** 1Faculty of Engineering Sciences, Heidelberg University, 69120 Heidelberg, Germany; 2Biosputnik LLC., New York, NY 10002, USA; 3Division of Pulmonary, Allergy, and Critical Care Medicine, Department of Medicine, University of Pennsylvania, Philadelphia, PA 19104, USA; 4Emeritus Professor of Organic Chemistry and Biochemistry, University Hamburg, 20148 Hamburg, Germany

**Keywords:** lymphangioleiomyomatosis, Nitazoxanide, lung disease, rare disease, drug repurposing

## Abstract

Lymphangioleiomyomatosis (LAM) is a rare genetic lung disease. Unfortunately, treatment with the mTORC1 inhibitor Rapamycin only slows disease progression, and incomplete responses are common. Thus, there remains an urgent need to identify new targets for the development of curative LAM treatments. Nitazoxanide (NTZ) is an orally bioavailable antiprotozoal small molecule drug approved for the treatment of diarrhea caused by *Giardia lamblia* or *Cryptosporidium parvum* in children and adults, with a demonstrated mTORC1 inhibitory effect in several human cell lines. NTZ’s excellent safety profile characterized by its more than 20 years of clinical use makes it a promising candidate for repurposing. Our rationale for this study was to further investigate NTZ’s effect using in vitro and in vivo LAM models and to elucidate the underlying molecular mechanism beyond mTORC1 inhibition. For this purpose, we investigated cell proliferation, cell viability, and changes in protein phosphorylation and expression in primary human cell cultures derived from LAM lung samples before translating our results into a syngeneic mouse model utilizing Tsc2-null cells. NTZ reduced cell growth for all tested cell lines at a dose of about 30 µM. Lower doses than that had no effect on cell viability, but doses above 45 µM lowered the viability by about 10 to 15% compared to control. Interestingly, our western blot revealed no inhibition of mTORC1 and only a mild effect on active ß-Catenin. Instead, NTZ had a pronounced effect on reducing pAkt. In the mouse model, prophylactic NTZ treatment via the intraperitoneal and oral routes had some effects on reducing lung lesions and improving body weight retention, but the results remain inconclusive.

## 1. Introduction

Tuberous sclerosis (TS), also called tuberous sclerosis complex (TSC), is a rare, multi-systemic genetic disease that causes benign tumors to grow in the brain and on other vital organs such as the kidneys, heart, eyes, lungs, and skin. Most cases of TS occur sporadically in the form of a de novo mutation, without any previous family history [1]. About one-third of adult women with TS develop lung lesions, which are much less commonly seen in men. Lung lesions include lymphangioleiomyomatosis (LAM) and multinodular multifocal pneumocyte hyperplasia [2]. LAM is a tumor-like disorder characterized by cystic lung lesions due to the proliferation of abnormal smooth muscle cells (LAM cells) in the lungs, leading to progressive shortness of breath and recurrent pneumothoraces. There are two manifestations of LAM, an inherited form and a sporadic form. However, contrary to TS, the inherited form, tuberous sclerosis complex LAM, is more common. The sporadic form only manifests after a spontaneous mutation in TSC2. TSC1 and TSC2 respectively encode for the hamartin (TSC1) and tuberin (TSC2) proteins, which together form the tuberous sclerosis complex. This complex inhibits mammalian target of Rapamycin 1 (mTORC1), a central regulator of cell growth and cell proliferation. Therefore, a loss of function mutation in either TSC1 or TSC2 leads to activation of the mTOR pathway, resulting in abnormal proliferation of the aforementioned LAM cells, as illustrated in Figure 1 [3,4].

The current treatment options for LAM include mTOR inhibition, oxygen therapy, procedures to remove air or fluid from the chest or abdominal area, procedures to remove angiomyolipoma (AML), or benign kidney tumors and lung transplantation. The first and only FDA-approved treatment for LAM is the immunosuppressant Rapamycin, which acts by inhibiting mTORC1. Rapamycin, while effective for many, has some clinical challenges. It stops lung function degradation and improves pulmonary and extra-pulmonary manifestations during treatment, but cessation of therapy leads to further disease progression. As it is categorized as FDA pregnancy risk category C, patients are encouraged to discontinue treatment 12 weeks before pregnancy, although data on Rapamycin treatment during pregnancy is sparse [5]. Song and colleagues reviewed other drugs currently in development for LAM, including autophagy inhibitors and compounds targeting tyrosine kinases [6]. Resveratrol, a naturally occurring polyphenol, was recently tested in a clinical phase 2 trial in combination with Rapamycin. While the therapy was safe and well tolerated, the primary efficacy endpoint was not met [7]. Thus, there is an unmet medical need to develop additional treatment options for LAM.

Nitazoxanide (NTZ) is an orally bioavailable antiprotozoal small molecule drug approved for the treatment of diarrhea caused by *Giardia lamblia* or *Cryptosporidium parvum* in patients 1 year of age and older. In protozoa and anaerobic bacteria, it interferes with the pyruvate:ferredoxin oxidoreductase (NQO1) enzyme-dependent electron transfer reaction, thereby impeding anaerobic energy metabolism [8,9]. In a 2012 study investigating the effect of NTZ on the intracellular proliferation of *Mycobacterium tuberculosis* [10], it was demonstrated that NTZ stimulates autophagy and inhibits mTORC1, thus preventing the intracellular proliferation of *M. tuberculosis*. The team also established NQO1 as the molecular target of NTZ and subsequently tested the effect on mTOR signaling in TSC2^+/+^ and TSC2^−/−^ mouse embryonic fibroblasts. Interestingly, NTZ and another NQO1 inhibitor both attenuated mTOR activity in TSC2^+/+^ cells, but not in the TSC2^−/−^ cells. Other research additionally solidified the hypothesis of NTZ as an inhibitor of autophagy through mTORC1 inhibition. Shou and colleagues investigated the effects of Tizoxanide, the active metabolite of NTZ, on autophagy using murine RAW264.7 macrophage cells. Measuring the activity of several autophagy-related pathways using western blotting, they concluded that Tizoxanide effectively induced the formation of autophagy vacuoles while simultaneously inhibiting PI3K, AKT, and mTOR activation. Pretreatment with Rapamycin enhanced those effects [11]. Due to the connection of LAM to dysfunctional mTORC1 signaling, the idea was formed to repurpose NTZ for the treatment of LAM. To evaluate the potential influence of NTZ on LAM disease, preliminary experiments using mouse embryonic fibroblasts and the LAM-related angiomyolipoma cell line 621-101 were performed, with the read-out being S6 phosphorylation as a stand-in for mTORC1 activity. In the presence of serum, 50 µM and 100 µM doses of NTZ were effective and reduced S6 phosphorylation to a level similar to that achieved by Rapamycin treatment. The observed effect was most pronounced for the wild-type cells. This strong effect, compared to the less pronounced results of two cell lines lacking TSC2, suggests a partially though not exclusively TSC2-dependent mechanism of action of NTZ (unpublished data), in line with what was observed by Lam and colleagues [10]. The improved adverse events profile of NTZ, while avoiding potential therapy cessation of Rapamycin, as well as a proven action in an in vitro disease model, make NTZ an intriguing candidate for repurposing with the ultimate goal of clinical use in lymphangioleiomyomatosis.

**Figure 1 biomolecules-14-01236-f001:**
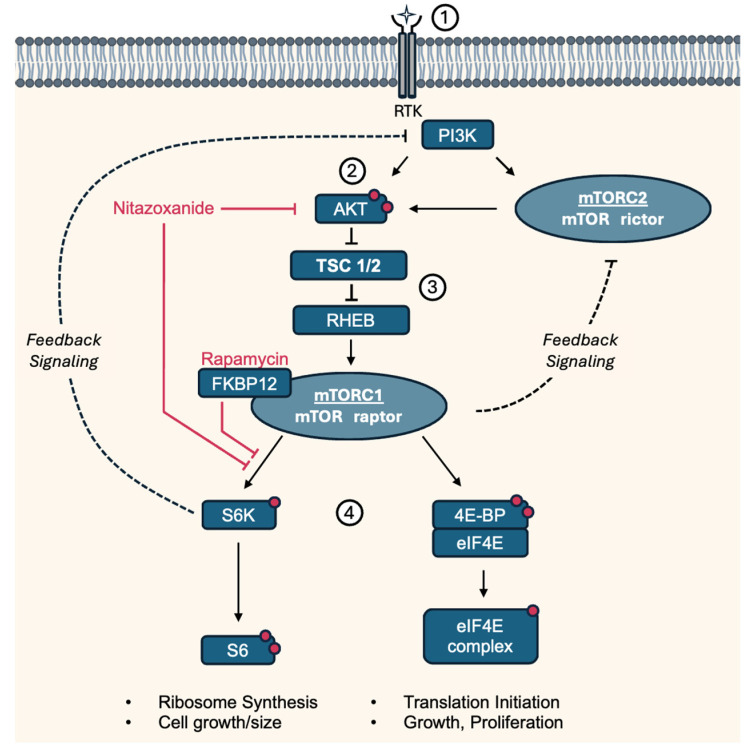
Simplified pathway and targets of Nitazoxanide and Rapamycin involved in LAM/TS. (1) Growth factors regulate mTORC1 activity through the PI3K-AKT pathway by activating PI3K, which leads to the production of PIP3 and the recruitment of AKT to the plasma membrane. (2) At the membrane, AKT is phosphorylated at two sites by PDK1 and mTORC2. (3) Activated Akt phosphorylates and inhibits TSC1/2, a negative regulator of mTORC1. TSC suppressor mutations in LAM lead to constitutively active mTORC1 through the RHEB GTPase, resulting in (4) downstream phosphorylation of S6K and 4E-BP. Rapamycin binds to cytosolic FKBP12 and thereby acts as an allosteric inhibitor of mTORC1, preferentially inhibiting S6K/S6 phosphorylation. mTORC1/S6K and mTORC2 mediate negative feedback of PI3K/AKT [12]. 4E-BP: Eukaryotic translation initiation factor 4E-binding protein 1, AKT: Protein kinase B, eIF4E: Eukaryotic translation initiation factor 4E, FKBP12: Protein FK506-binding protein 12, mTORC: Mammalian target of Rapamycin complex, PDK1: Phosphoinositide-dependent kinase-1, PI3K: Phosphatidylinositol 3-kinase, PIP3: Phosphatidylinositol (3,4,5)-trisphosphate, Raptor: Regulatory-associated protein of mTOR, RHEB: Ras homolog enriched in brain, Rictor: Rapamycin-insensitive companion of mammalian target of rapamycin, RTK: Receptor tyrosine kinase, S6K: P70-S6 Kinase 1, S6: Ribosomal protein S6, TSC1/2: Tuberous sclerosis complex. Red dots indicate activating phosphorylation sites.

Here, we present the results of our investigation into NTZ as a potential treatment for LAM, utilizing established in vitro and, for the first time, in vivo models of LAM. Following confirmation of cell growth inhibition in vitro, we proceeded to elucidate NTZ’s effects on various pathways via western blot analysis before transitioning to a mouse model. Employing subcutaneous and tail vein tumor cell injection, we assessed different doses and routes of administration, investigating NTZ’s potential for standalone use or combination treatment with Rapamycin. While the findings show promise, certain aspects remain inconclusive, underscoring the need for further research to fully explore its therapeutic potential.

## 2. Materials and Methods

### 2.1. Compounds and Reagents

Nitazoxanide (Cat. No.: HY-B0217) and Rapamycin (Cat. No.: HY-10219) were both obtained from MedChemExpress LLC. (Monmouth Junction, NJ, USA). Stocks of 1–100 mM were made in DMSO (Sigma-Aldrich, St. Louis, MO, USA) and stored in aliquots at −20 °C, prior to appropriate dilution for inhibition studies. All other reagents were of the highest grade possible.

### 2.2. In Vitro Cell Culture

Human LAM patient-derived angiomyolipoma (AML) 621-102 TSC2-null cells were derived from a sporadic LAM-associated renal AML carrying a biallelic inactivation of *TSC2* [13]; these were provided by Dr. Elisabeth Henske, Brigham and Women’s Hospital and Harvard Medical School, Boston, MA. Lung tissue was obtained from LAM patients under the National Disease Research Interchange (NDRI, Philadelphia, PA, USA)-approved protocols, and LAM lung cell lines were derived, as previously described in [14,15]. Mouse TSC2-null TTJ cells were obtained as described in [16]. Cell lines were grown in DMEM media (Corning, Manassas, VA, USA) supplemented with 100 U/mL penicillin, 100 μg/mL streptomycin (Gibco, Gaithersburg, MD, USA), 10 mM glutamine, and 10% fetal bovine serum (FBS) (Atlanta Biologicals, Atlanta, GA, USA) and were maintained at 37 °C in a humidified atmosphere with 5% CO_2_.

### 2.3. Cell Growth Assay

Cells were plated at 5 × 10^3^ cells/well in a 96-well plate in 200 µL of 2.5% FBS DMEM media. Two identical plates were prepared for each set of conditions—one plate to be imaged at 48 h post drug addition and the other plate at 72 h post drug addition. One additional plate was prepared with four lanes to be imaged 24 h after cell plating (used as a baseline for later percent-inhibition calculations). To help cells evenly settle in pates, the plates were left at room temperature for one hour before being placed in the incubator at 37 °C, 5% CO_2_ for 24 h. Cells were dosed with drug compounds at t = 24 h. For processing and read-out, 20 μL of 25% glutaraldehyde was added to each media-containing well, the cells were allowed to fix for 15 min, and then they were gently agitated every few minutes by hand before tapping out the liquid from wells onto paper towels. Next, 50 μL of 0.05% crystal violet (C3886, Sigma-Aldrich, St. Louis, MO, USA) in 25% methanol was added to each well, and the plate was placed on a shaker (60 rpm) at room temperature for 15 min. To wash away excess stain, whole plates were dunked in a large container of water (repeated twice or until the water ran clear) and tapped dry on paper towels before air drying for at least 1 h. Then, 200 μL of 100% methanol was added to each well and mixed on a shaker ~10 min before being read at A590 nm. The readings were averaged for each set of 4 technical replicate wells, and the %-inhibition was determined from test plate controls.

### 2.4. Trypan Blue Viability Assay

The method was adapted from “Cell Viability Analysis Using Trypan Blue: Manual and Auto-mated Methods” by Kristine S. Louis and Andre C. Siegel [17]. Cells were seeded on 6-well plates at t = 0 at around 8 × 10^5^ cells/well. Drugs were added in duplicate at 24 h post-seeding. Trypan blue staining/cell counting was performed at 48 h post-seeding (24 h post-drug treatment). The cells were trypsinized, spun down into a pellet, and resuspend in 1 mL of media. Next, 10 µL of 0.4% trypan blue dye was combined with 10 µL of cell suspension, and the mixture was left to sit at room temperature for 5 min. All viable and non-viable cells were counted (dead cells stained dark blue), and the % viability was determined as follows: % viable cells = (# viable cells)/(total cells present).

### 2.5. Western Blot

NHLF and LAM cells (biological triplicates) were seeded in 6-well cell culture plates at 5 × 10^5^ total cells per well in 2 mL of DMEM with 10% fetal calf serum and incubated in a 37 °C humidified incubator in 5% CO_2_ overnight. The next day, the fully adherent cells were washed once with prewarmed DPBS and serum starved by the addition of 2 mL of DMEM 0.1% BSA medium and incubated for 2 h prior to compound addition. Control (final 0.1% DMSO in 0.1% BSA medium), Nitazoxanide (30 µM final concentration), or Rapamycin (10 nM final concentration) additions were made in triplicate wells for each compound, and the incubation at 37 °C continued for 18–20 h. The plates were then placed on ice; the cells were washed with 2 mL of ice-cold DPBS; 100 µL of RIPA lysis buffer containing PMSF (1 mM), PhosSTOP (Roche under Sigma-Aldrich, St. Louis, MO, USA) phosphatase, and cOmplete (Roche under Sigma-Aldrich, St. Louis, MO, USA) protease inhibitors was added; and the cells were held on ice for 30 min. The lysed cell supernatant was scraped off into 1.5 mL Eppendorf tubes and centrifuged at 10,000× *g* for 20 min at 4 °C to remove large cell debris. The protein concentrations in supernatants were determined by the standard BCA (Pierce) assay, and samples were denatured at 90 °C for 5 min in PAGE sample buffer containing 0.1 M DTT. Then, equal amounts of protein were added to the wells of 15-well 1 mm 10–20% gradient NOVEX polyacrylamide gels. Proteins were separated in Tris-glycine SDS buffer for ~1 h at a constant 165 V current at room temperature. The gels were transferred by semi dry blot to nitrocellulose (7 min P zero setting iBlot), blocked for 45 min in TBS LiCor blocking buffer, and transferred to a four-primary-antibody mixture (anti-pS6 Ser235/236, p4EBP1 Th37/46, anti-pAkt Ser473, and anti-Active ß-Catenin, all at 1:1000 dilution, and anti-ß-Actin at 1:10,000 dilution; all antibodies were from Cell Signaling, Danvers, MA, USA) in TBS blocking buffer, 0.1% Tween20 overnight at 4 °C with gentle shaking. Immunoblots were then washed 4 × 5 min in TBS containing 0.5% Tween20 (TTBS) and incubated for 1 h with fluorescent secondary antibody (LiCor 800W donkey anti-rabbit) at 1:15,000 dilution in TBS blocker containing 0.2% Tween20. Immunoblots were again washed 4 × 5 min with TTBS, 1× TBS, then dried at 37 °C for 10 min, and fluorescent band signals were determined on a LiCor Odyssey imager (LiCor, Lincoln, NE, USA).

### 2.6. Experimental Animals

All experiments involving animals conformed to the Guide for the Care and Use of Laboratory Animals published by the US National Institutes of Health (NIH Publication eighth edition, update 2011) and were approved by the Institutional Animal Care and Use Committee of the University of Pennsylvania. The studies were carried out in compliance with all ethical regulations. Mice were kept and observed by professional husbandry staff in the CRB vivarium. Rooms ranged from 68 to 78 °F and 20–70% humidity. The lights were on a 12 h cycle on at 7 a.m. and off at 7 p.m. year around. Eight- to ten-week-old NCRNU-F athymic nude female mice (Taconic, Germantown, NY, USA) were used for all in vivo portions of this project.

### 2.7. Preparation of Mouse Lungs

Mice were euthanized by a single dose of Euthanasia Solution (Pentobarbital based). The chest cavity was exposed, and the lungs were cleared of blood by perfusion with cold PBS via the right ventricle. The lungs were inflated from control and experimental animals at a constant 25 cm H_2_O pressure, measured from the animal’s chest. Two percent paraformaldehyde was used as the fixative. After inflation, the lungs were carefully dissected out and placed in a container full of 2% paraformaldehyde for fixation overnight at 4 °C. The lungs then went through a dehydration process at 4 °C with gentle rotation. The first step was 4 × 30 min PBS washes. Next, the lungs were placed in 30% ethanol for 2 h and then 50% ethanol for 2 more hours. The lungs were then left in 70% ethanol overnight. The next day, the lungs were changed to 95% ethanol and left overnight. On the third day of lung dehydration, the lungs were placed in 100% ethanol and left overnight. The next morning, the lungs were placed in fresh 100% ethanol and stored at −20 °C for processing by the Children’s Hospital of Philadelphia Pathology Core.

### 2.8. In Vivo Treatments

Subcutaneous pilot study: NCRNU-F athymic nude female mice received a subcutaneous injection of 5 × 10^6^ Tsc2-null TTJ cells into one flank on day one. They were treated with either NTZ 50 mg/kg intraperitoneally (i.p.) daily, NTZ 500 mg/kg orally in chow (Animal Specialties and Provisions, Quakertown, PA, USA), or Rapamycin 4 mg/kg i.p. every other day. Treatment was started either on day 3 (prophylactic regimen) or upon the tumor reaching 5 mm either in length or width (therapeutic regimen). Tumor width and length were measured every 3 days, and the volume was calculated according to the formula V = length × width^2^ × 0.5. Mice were sacrificed on day 28 after injection of cells or upon experiencing more than 20% loss of body weight.

Tail vein study: NCRNU-F athymic nude female mice received 800,000 Tsc2-null TTJ cells injected into the tail vein. Based on the two most promising treatment regimens in the subcutaneous study, they received NTZ 500 mg/kg orally in chow (Animal Specialties and Provisions) daily, NTZ 100 mg/kg intraperitoneally daily, or a combination regimen with Rapamycin. Rapamycin was used as a control at either 0.1 mg/kg i.p. or 0.5 mg/kg i.p. every other day.

### 2.9. Data Analysis

Statistical analysis was carried out using the R Analytics Platform (Version 1.4.1106).

Cell Proliferation: In order to select an adequate statistical test, the Shapiro–Wilk test was applied to check for normality of the data, and the Bartlett test was performed to determine homoscedasticity (*p*-value < 0.05). Depending on whether the conditions for parametric tests were met, One-Way ANOVA (for parametric data) or the Kruskal–Wallis test (for nonparametric data) was used to calculate statistically significant differences between groups, followed by post hoc analysis using Dunnett’s test for comparison to the negative control with Holm–Bonferroni adjustment for multiple testing.

IC50: IC50 analysis was performed using the ‘drc’ R package [18], by applying the four-parameter log-logistic function.

Viability: Since for each viability analysis, only 2 or 3 datapoints were available, no statistical analysis was applied. The small sample size was not sufficient to yield a satisfactory power value.

Western Blot: The density of specific bands was determined and normalized to β-Actin in each lane. Graphs of immunoblots were made with PrismGraph Version 9 (San Diego, CA, USA) as each specific protein, with the control value set as one, relative to β-Actin in each lane, and the data are expressed +/– SEM. The significance was determined by Student’s *t* test, where * *p* < 0.05 and NS indicates a non-significant difference.

Subcutaneous Study: Tumor size was determined as previously described. *p*-Values were calculated using pairwise two sample *t*-tests with Benjamini–Hochberg adjustment and plotted using the R ggboxplot function of the ggpubr package version 0.6.0.

Tail Vein Study: The Dunnett test was conducted to determine the significance of percent-lesion group differences with Benjamini–Hochberg adjustment.

Lesion Calculation: After mouse lung samples were processed as described above and submitted to the Children’s Hospital of Philadelphia Pathology Core, slides were cut from each sample, and one slide per sample was stained with hematoxylin and eosin for morphometric analysis. The H&E slide was then scanned at 20× magnification and stitched into a single image by the Pathology Core using Aperio ScanScope CS-O (Aperio Technologies, Inc., Vista, CA, USA). Images were then analyzed in Photoshop. First, the white space surrounding and between the lung lobes was removed. Next, areas with lesions present were carefully selected via the quick selection tool and copied to their own layer. Any additional parenchyma or other lung structures selected by the quick select tool were removed from the lesions layer. The percent-lesion was then calculated by dividing the number of pixels present in the lesion layer by the number of pixels present in the total lung layer.

## 3. Results

### 3.1. NTZ Inhibits Cell Growth and Affects Viability in LAM-Related Cells

As a first step, we tested how well incubation with Nitazoxanide was able to inhibit cell growth by exposing different LAM-related cells to NTZ for either 48 h or 72 h. Two of the three tested sets had verifiable TSC2 mutations–the murine TTJ and human angiomyolipoma 621 cells–which allowed us to compare a TSC2-inactive (TSC2-) to a TSC2-active (Tsc2+) state. For the primary human-derived cells LAMD100 and LAMHUP, the TSC2 status is unknown, and the normal human lung fibroblast line AHHP148 was used as a control. NTZ was able to inhibit proliferation for all tested cell lines in a dose-dependent manner. After determining the % proliferation from the raw values, we calculated an IC50 (Table 1) for all the cell lines at 48 h and 72 h incubation times using the R drc package, version 3.0-1.

The resulting curves for the 48 h incubation are visualized in Figure 2a. The data looked similar at 72 h and is not included for brevity. Across all three sets, the TSC2+ cells were more sensitive to NTZ. The AML cell line specifically was the most sensitive, followed by TTJ and then the primary human-derived cells. Since in some extreme cases, we observed cells dying after NTZ treatment, we next aimed to investigate the compound’s effect on general viability. Figure 2b shows the normalized results for the viability assessment of all cell lines. Due to the small number of replicates, statistical analysis was omitted for this section. Nevertheless, there are certain trends to be observed. NTZ at its predicted IC50 value of around 27 μM in the human-derived cells does not seem to negatively affect viability in any of the tested cell lines. However, a higher dose of 45 μM did lower the viability by about 10–15% compared to the DMSO-only control. The primary normal human lung fibroblasts seem to be affected the strongest, with their viability at 45 μM reduced by about 23%.

### 3.2. NTZ Has a Partial Inhibitory Effect on the Canonical Wnt ß−Catenin Growth Pathway

To expand our understanding of NTZ’s pathway modulatory effects, we performed western blotting experiments using TSC2-null TTJ cells before beginning the in vivo experiments to better understand how the TTJ cells would respond to NTZ treatment. NTZ at 30 μM, a dose approximately double the predicted IC50 for TTJ-TSC2-null cell growth inhibition, showed a trend for inhibition of about 50% of mTORC1-driven pS6 (Ser235/236) and, similar to Rapamycin, did not inhibit p4E-BP1 (Thr37/46). Interestingly, unlike the increase in pAKT (Ser473) by Rapamycin, NTZ decreased pAKT by about 75% and active β-Catenin by about 50% (Figure 3). Rapamycin at 10 nM, the clinical trough concentration of Rapamycin at steady state after a 2 mg dose, inhibited 100% of the phosphorylation of pS6 (Ser235/236) but only 0–20% of p4E-BP1 (Thr37/46). Rapamycin also increased the phosphorylation of pAKT in our experiment, which aligns with its known effect on an mTORC1/mTORC2 feedback loop [19,20].

### 3.3. NTZ Inhibits Tumor Growth in a Subcutaneous Mouse Model of LAM

As a next step, we sought to translate the observed results into an in vivo experiment. We began with a subcutaneous model for its direct measurability, albeit with limited clinical translatability. Our aim encompassed exploring various initiation points for treatment, considering the potential for earlier intervention with the advent of improved LAM diagnosis. Prophylactic treatment was started on day 3 post tumor cell injection for both NTZ groups, while therapeutic intervention was initiated upon the tumor reaching 5 mm in either width or length for the Rapamycin control. The results are shown in Figure 4. As anticipated, the onset of tumor growth in the untreated group was prompt, progressing steadily throughout the experiment’s duration. The positive control of Rapamycin at 4 mg/kg, a relatively high dosage, exhibited robust efficacy. Both treatments involving NTZ demonstrated promising outcomes, maintaining comparability with each other until day 22. At this point, the i.p. group showed a notable escalation in tumor size, contrasting with the sustained statistically significant efficacy of the oral administration, although to a lesser degree than Rapamycin. Notably, the oral regimen also exhibited a more consistent effect on halting tumor growth, as evidenced by the lower variability compared to the i.p. treatment group.

### 3.4. NTZ Has Positive Effects on %-Lesions, Body Weight, and Survival in the Tail Vein Model of LAM

To enhance the clinical relevance of our findings, we proceeded to assess various concentrations of NTZ, along with a combined treatment incorporating a lower dose of Rapamycin, in the tail vein model of LAM. Injection of Tsc2-null tumor cells into the tail vein results in the growth of multiple lesions in the lungs, characterized by the expression of vascular endothelial growth factor D and the promotion of lymphangiogenesis [20]. Figure 5 depicts the outcomes of this experiment. Tail vein injection of Tsc2-null cells resulted in a notable decline in body weight among the untreated control group. As expected, naïve mice exhibited weight gain, a trend also observed in animals treated with high-dose Rapamycin, either alone or in combination with NTZ. Although minor fluctuations were noted in other groups, they generally maintained their initial weight by the study’s conclusion (Figure 5c).

Unsurprisingly, the vehicle-treated group also displayed the highest lesion development, which was significantly attenuated by high-dose Rapamycin treatment (Figure 5b). However, the addition of NTZ to high-dose Rapamycin did not yield further improvements in lesion reduction, potentially due to the robust efficacy of Rapamycin alone, making comparative evaluation challenging. Consequently, we explored a lower dosage regimen, which, as anticipated, demonstrated inferior performance compared to the 0.5 mg/kg group. Combining 0.1 mg/kg Rapamycin with 100 mg/kg NTZ, both given i.p., increased the efficacy by approximately 10%; however, the results were akin to administering NTZ intraperitoneally at the same dose alone. Doubling the dose while transitioning from subcutaneous to tail vein cell injection appeared to enhance the efficacy of intraperitoneal NTZ, even slightly exceeding the 500 mg/kg NTZ oral regimen in %-lesion reduction.

## 4. Discussion

Here, we report a first investigation into antiparasitic Nitazoxanide’s potential as a novel treatment for the rare disease lymphangioleiomyomatosis. Our in vitro experiments yielded tentative positive results, with Nitazoxanide effectively inhibiting the tested cell lines, albeit exhibiting a less-pronounced performance in primary human-derived cells. This outcome was somewhat anticipated, considering the inherent heterogeneity of these primary cells compared to the established cell lines. Unfortunately, for all three tested cases, the control with functional TSC1/2 reacted more strongly to Nitazoxanide, something that is undesirable in a clinical context where the effect on healthy cells should be minimized. This is further illustrated by the effect on overall viability, which was worst for the normal human lung fibroblasts, showing a dose-dependent reduction in viability with increasing dose. The statistical significance of this result could not be analyzed due to the small sample size (n = 3 per cell line), a shortcoming of this analysis. An IC50 would ideally be in the nanomolar range, which we did not achieve with our experiments. Further studies into NTZ’s exact mechanism and how it relates to LAM will have to be performed to gain a better understanding and eventually overcome this drawback. Luckily, the compound has recently gathered more interest due to its potential efficacy in treating COVID-19. There has been a lot of investigation into its anti-inflammatory effects [21,22], influence on PDI-mediated oxidoreductase mechanisms [23], autophagy induction [10,11,24], and others [25]. Our own western blot analysis provided some confirmatory mechanistic insights in new cell lines, revealing effects for further exploration in LAM. In Tsc2-null TTJ cells, in which both mTORC1 and Wnt ß-Catenin activation have been shown to contribute to excessive growth [26], NTZ partially inhibits both growth pathways in vitro. While Wnt ß-Catenin inhibition by Nitazoxanide has been published before in cancer [27], the 75% reduction in pAkt is a considerable effect in LAM specifically, as the approved drug Rapamycin increases pAkt (Ser473) and active ß-Catenin.

There is experimental evidence that activation of PI3K/Akt signaling through insulin receptors might be an additional important growth stimulus in LAM [28,29]. Akt is a relevant target, as it phosphorylates and inhibits the TSC1/2 complex, further amplifying mTORC1 activity in LAM cells (see Figure 1). Activation of Akt is a multi-step process. In short, phosphoinositide-dependent kinase-1 (PDK1) phosphorylates Akt on the Thr308 residue, only partially activating it. Full activation of Akt occurs when it is phosphorylated on the Ser473 residue by mTORC2. As is the case with many signaling pathways, there exists a negative feedback loop to counteract impeded mTORC1 signaling, which becomes activated upon treatment with Rapamycin. This happens mainly via insulin/PI3K/Akt signaling and has been observed in the form of increased Akt in biopsies of cancer patients after Rapamycin treatment [30,31]. There is an additional mechanism through which mTORC1 regulates mTORC2, with insulin receptor-binding protein Grb-IR as the main actor [32]. This effect is seen as a serious drawback of rapalogs and has prompted the development of alternative mTOR inhibitors targeting both complexes, as mTORC2 is Rapamycin-insensitive [12]. Dual PI3K/mTOR inhibitors might also be able to overcome the resistance mediated by feedback loops, and the fact that NTZ is able to inhibit Akt phosphorylation is very promising.

Consequently, Rapamycin and NTZ might have additive effects on Tsc2-null TTJ cell growth inhibition. However, it is imperative to validate these findings by extending our investigations to include additional cell lines such as AML or primary human-derived cells. Thus far, we have only investigated Akt phosphorylation at a single site (Ser473). As the described signaling networks are incredibly complex, additional mechanistic studies will be of utmost importance for the rational design of a potential combination therapy.

Next, we performed a first proof-of-concept in vivo study utilizing a subcutaneous model of LAM, wherein cells are directly injected under the skin, making the tumor size measurable. While ultimately NTZ is to be administered orally, we chose to investigate intraperitoneal injection as well. The dosing of NTZ is not straightforward; it only exhibits oral bioavailability of about 30%, which can increase up to 50% with food [8,33]. This low value and poor lung concentrations, discovered while investigating its COVID-19 efficacy, are major challenges that need to be addressed for LAM as well. However, in the subcutaneous model, 500 mg/kg administered via chow proved to be effective in reducing tumor size, more so than the 50 mg/kg intraperitoneal injection. It needs to be noted that we employed a prophylactic regimen, given the ongoing research and discovery of novel diagnostic and prognostic biomarkers in LAM, making an earlier diagnosis and intervention possible in the future [34].

We opted to test both administration routes in the more complex and clinically relevant tail vein cell injection model of LAM, expanding the groups by a low-dose Rapamycin regimen and a combination treatment. We also doubled the i.p. dose to 100 mg/kg NTZ, which is still considered safe according to the NTZ FDA safety review [35]. While all groups showed some level of lesion reduction, high-dose Rapamycin performed the best by far. Combining NTZ at 100 mg/kg administered intraperitoneally with a low dose of Rapamycin enhanced its efficacy. However, this combination was less effective compared to a high dose of Rapamycin and showed no significant difference from administering NTZ alone, either at 100 mg/kg intraperitoneally or 500 mg/kg orally. Still, it could be interesting to consider future investigations into potential synergies with Rapamycin, even at a higher dose of NTZ, given its well-studied safety profile. Rapamycin, despite its efficacy, has the disadvantages of notable side effects and a significant portion of non-responders. It also does not cure LAM but only inhibits disease progression; thus, stopping the treatment can lead to a worsening of symptoms. Exploring the prospect of dose reduction through combination therapy with a safer compound such as NTZ could prove to be a promising strategy, as low-dose Rapamycin has been associated with fewer side effects [34,36]. The idea of a combination treatment is not new in LAM, as in 2015, Alayev and colleagues reported promising results from combining Rapamycin with resveratrol, an autophagy inhibitor, in a Tsc2-null xenograft tumor model. They were able to show that this combination inhibited PI3K/Akt/mTORC1 signaling, leading to apoptosis [37].

The study has several limitations that impact the interpretation of its findings. First, the viability measurements lack statistical support, which raises concerns about the reliability of the data, although similar effects in other cancer cells have been observed by others at comparable doses [38]. Additionally, the western blot data were only generated in murine TTJ cells, limiting the generalizability of the findings to other cell types or species. The subcutaneous model used is less clinically relevant than the tail vein injection model, which better mimics metastatic spread. Nonetheless, this being a proof-of-concept study, we decided to use this model as a pilot before moving into the more complex tail vein model. Moreover, the prophylactic treatment explored is not yet translatable to clinical practice, and the tail vein model data were not validated through immunohistochemistry, leaving gaps in confirming tissue-specific effects. The study also lacks clear mechanistic insights, as the in vitro findings were not sufficiently corroborated by the in vivo effects, and the dose and route of administration remain to be optimized for potential clinical application. Ultimately, this first investigation into Nitazoxanide’s potential for treating LAM gave us some interesting insights but left us with some unanswered questions. Experiments need to be repeated, additional cell lines should be tested, and studies need to be adequately statistically powered to gain a better understanding. As the effects observed so far have only been of medium strength, dose optimization and treatment schedule need to be considered and potentially optimized.

## Figures and Tables

**Figure 2 biomolecules-14-01236-f002:**
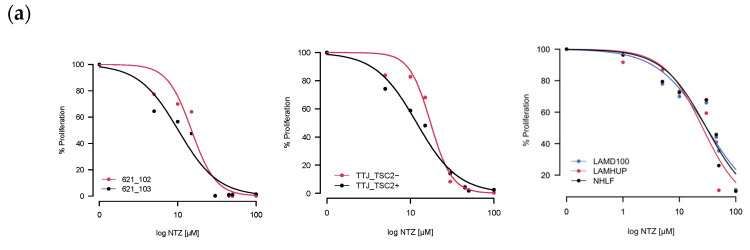
NTZ affects cell proliferation and viability in LAM-related cell lines. (**a**) Percent proliferation for different LAM-related cell lines: murine-derived TTJ cells; human angiomyolipoma cell line 621; primary human-derived LAM cells. Curves were created by applying the four-parameter log-logistic function. (**b**) Percent viability for the same cells. Trypan blue staining/cell counting was performed at 48 h post-seeding. Assays were performed in triplicate.

**Figure 3 biomolecules-14-01236-f003:**
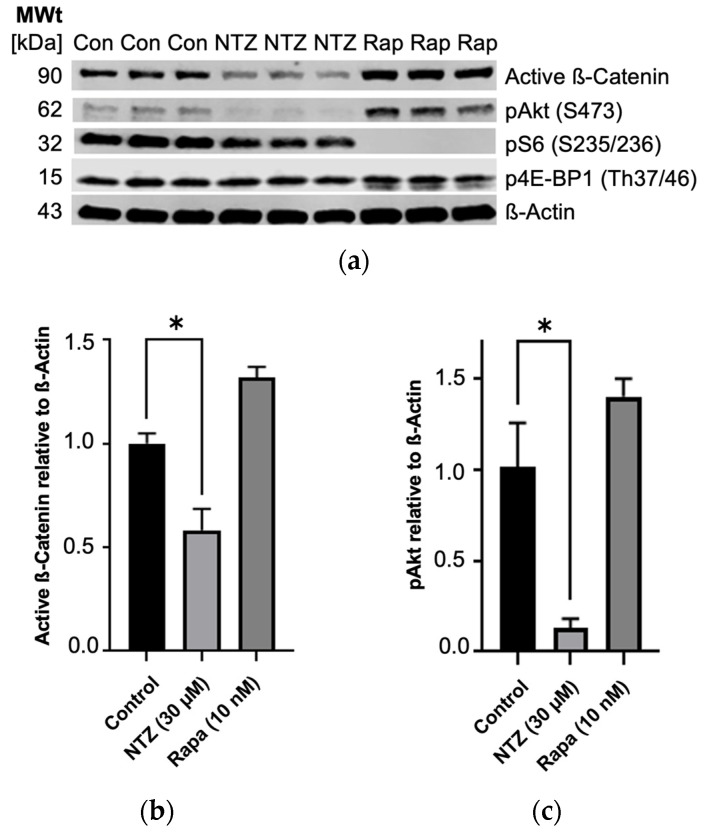
Partial inhibition of mTORC1 and Wnt ß-Catenin growth pathways in TTJ Tsc2-null cells by NTZ as determined by immunoblot analysis. (**a**) Immunoblot bands. (**b**) Active ß-Catenin is inhibited by NTZ. (**c**) pAkt is inhibited by NTZ and increased by Rapamycin (* *p* < 0.05). Original figures can be found in Appendix A.

**Figure 4 biomolecules-14-01236-f004:**
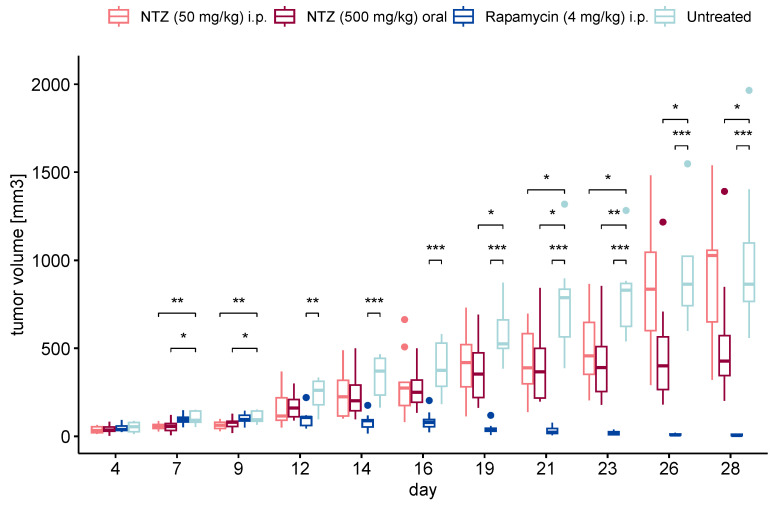
Tumor volume in the subcutaneous experiment. NTZ given intraperitoneally (n = 10) and orally formulated as chow (n = 10) on day 3 post tumor cell injection were able to significantly reduce tumor growth compared to the untreated control (n = 9). The positive control Rapamycin (n = 10) performed very well. (* *p* < 0.05, ** *p* < 0.01, *** *p* < 0.001). Dots represent outliers of the boxplot.

**Figure 5 biomolecules-14-01236-f005:**
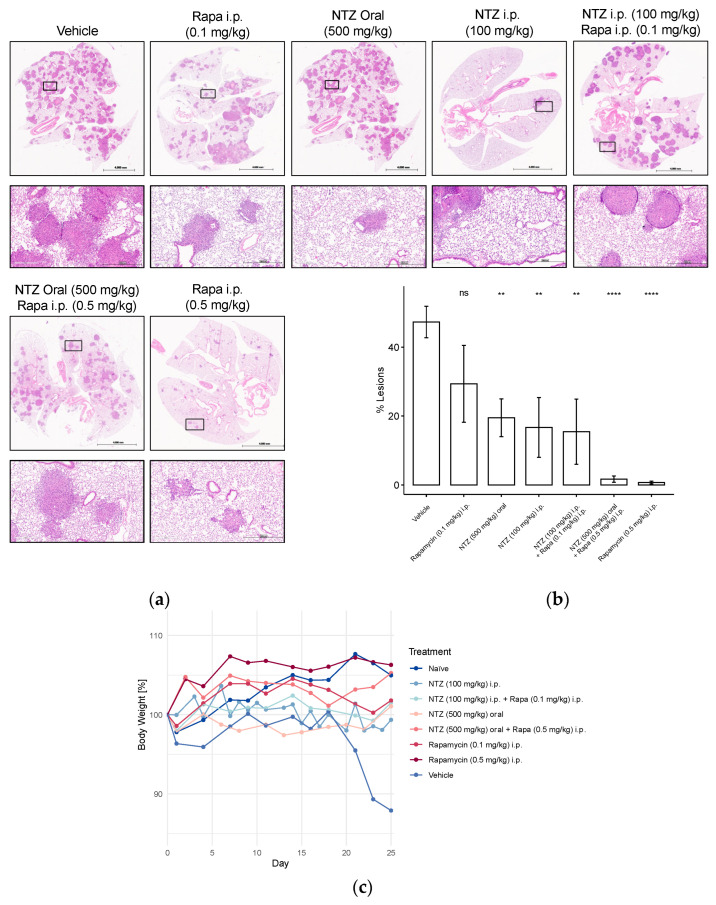
Representative H&E stainings, percent-lesions, and body weight changes in Tsc2-null lung lesions in a metastatic immunocompromised mouse model of LAM. Mice received 800,000 Tsc2-null TTJ cells injected into the tail vein. The animals received pharmacological treatment beginning 3–7 days after the injection of cells. Animals were euthanized 28 days after injection of the Tsc2-null cells. In two subsequent studies with a similar design, different NTZ and Rapamycin doses and combination were tested. (**a**) Lungs were inflated and stained with H&E. Scale bars: 4 mm and 500 μm. (**b**) Percent-lesion formation in lungs of the treated mice at takedown. Shown is an average of all mice in the treatment groups. (**c**) Development of body weight over the course of the study, presented as percentage change from baseline body weight on day 0. Group sizes were as follows: Vehicle n = 5, Rapa (0.1 mg/kg) i.p. n = 5, NTZ (500 mg/kg) oral n = 10, NTZ (100 mg/kg) i.p. + Rapa (0.1 mg/kg) i.p. n = 5, NTZ (100 mg/kg) i.p. n = 5, NTZ (500 mg/kg) oral + Rapa (0.5 mg/kg) i.p. n = 10, Rapa (0.5 mg/kg) i.p. n = 10. (ns non-significant, ** *p* < 0.01, **** *p* < 0.0001).

**Table 1 biomolecules-14-01236-t001:** Cells and their IC50 values as determined by a log-logistic regression model. *** *p* < 0.001, ** *p* < 0.01.

Cells		IC 50 [µM]
Human Angiomyolipoma Cells	621_103 TSC2+	10.18 ***
	621_102 TSC2-	15.01 ***
Murine BL6/TTJ Cells	TTJ TSC2+	11.91 ***
	TTJ TSC2-	17.46 ***
Primary Human-Derived LAM Cells	NHLF AHHP148	28.19 **
	LAMD100	29.13 **
	LAMHUP	23.96 ***

## Data Availability

The raw data supporting the conclusions of this article can be found in the Appendix A.

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
