# Peer review of "Repurposing Nitazoxanide for Potential Treatment of Rare Disease Lymphangioleiomyomatosis"

_biomolecules, 2024, doi:10.3390/biom14101236_

Round 1

Reviewer 1 Report

Comments and Suggestions for Authors

Bähr et al. in their study (Repurposing Nitazoxanide for Potential Treatment of Rare Disease Lymphangioleiomyomatosis) conducted a systematic examination into the effects of Nitazoxanide (NTZ) and Rapamycin on lung defects associated with Lymphangioleiomyomatosis (LAM), utilizing both in vitro and in vivo models. Their results effectively demonstrate that NTZ induces the downregulation of phosphorylated Akt without impacting mTORC1, which correlates with a reduction in lung lesions and improved body weight in the models tested.

However, it is crucial to show that while Tuberous Sclerosis (TS) and LAM are related, they remain distinct diseases. Given that the study title and experimental models specifically focus on LAM, the implications, and discussions should be restricted to LAM and not extrapolate to TS. To ensure clarity and precision, I strongly recommend revising the Abstract, Introduction, and Discussion sections to maintain this focus. 

In addition to the above suggestions, would it be beneficial to include a comprehensive table listing all existing treatments explored for LAM? Such a table could serve as a valuable reference point in the Introduction or other relevant sections of the paper, enhancing our understanding of the therapeutic backup.

Regarding Figure 1, the legend should be expanded to clearly define all abbreviations used  under the figure ensuring that the figure is easily interpretable.

Section 3.3 presents intriguing data on the impact of NTZ on tumor growth in a subcutaneous mouse model of LAM. However, to strengthen the manuscript, it is strongly recommended to include representative tumor images that depict the observed differences in size. Such visual evidence would significantly reinforce the findings of this paper.

The study concludes that while this is the first investigation into the potential of NTZ for treating LAM, many questions remain unanswered, and further studies with adequate power are necessary. This conclusion, however, risks dampening the enthusiasm for the potential of NTZ. Could the authors consider proposing a conceptual figure or model that outlines possible mechanistic insights into how NTZ might alter LAM disease progression? Such a proposal, based on the current in vitro and in vivo findings, would offer a more forward-looking perspective and stimulate further research in this area.

Lastly, what are the implications of the effects of NTZ on phosphorylated Akt in the context of LAM? Could this pathway be explored further to identify potential biomarkers or synergistic therapeutic strategies? 

Reviewer 2 Report

Comments and Suggestions for Authors This article investigates the potential of Nitazoxanide (NTZ) as a novel treatment for Lymphangioleiomyomatosis (LAM) using both cell models and in vivo animal models. Given the current lack of effective therapies for LAM, this study is of significant value in exploring new therapeutic avenues. However the study primarily conducted preliminary observations of the drug intervention effects, lacking in-depth mechanism analysis. The data presentation have some serious flaws As such, this manuscript is not suitable for publication and should be rejected   Major points 
  1. The title "Result 3.4: NTZ Inhibits Tumor Growth in the Subcutaneous Mouse Model of LAM" is misleading, as the actual model employed was the "tail vein model of LAM."
  2. The study's tail vein model of LAM lacks histological or morphological assessments before and after drug intervention, weakening the credibility of the results.
  3. While the in vitro studies on NTZ intervention identified differential signaling molecules, these findings are not supported by in vivo validation within the LAM model. Furthermore, the discussion section fails to provide a thorough analysis of the changes in these pathways, limiting a comprehensive understanding of how NTZ might modulate LAM's pathophysiology.
  4. Reference 19 is incorrectly cited; it refers to NTZ's intervention research in COVID-19, not LAM.
Minor Points:
  1. In Figure 4, the colors representing NTZ (50 mg/kg i.p.) and NTZ (500 mg/kg oral) are too similar, making them difficult to distinguish. Please revise.
  2. The study's limitations should be addressed and discussed in the "Discussion" section.

Round 2

Reviewer 1 Report

Comments and Suggestions for Authors

Dear Author, 

Thank you for addressing most of my comments in the revised version of your manuscript. Your efforts are evident, and the revisions have substantially improved the clarity of the paper.

Figure 1 is a significant strength of the manuscript, and you have not followed my suggestions properly.  Some of the abbreviations indicated in the legend are absent in the figure.

To enhance the comprehensibility of Figure 1 for all readers, please include definitions for all abbreviations within the figure legend. For example, PI3K should be defined as Phosphatidylinositol 3-kinase. Additionally, please clarify the activated states of AKT, 4E-BP, S6, and eIF4E in the legend. 

Furthermore, it would be helpful to update the figure to include information about Rapamycin and PI3K/Akt[12]. Currently, readers must refer to the indicated reference to fully understand the figure, which is inconvenient. Incorporating these details directly into the figure will make it more comprehensive and user-friendly.

It is strongly recommended to indicate stepwise reactions by adding steps 1,2,3……………..

Author Response

Dear Reviewer 1,

thank you for your comments. Especially the suggestion of including a more detailed figure in round 1 has really helped us to improve this manuscript. We hope that the further improvements now have made it clearer. The changes were marked in green this time, as to better separate them against the red changes from round 1.

Reviewer 2 Report

Comments and Suggestions for Authors

The author has carefully revised the manuscript according to the reviewers' comments, adding key images and data, which has significantly improved the quality of the paper. It is recommended for acceptance.

Author Response

Dear Reviewer 2,

thank you again for your comments in round 1, they were very helpful and we are happy that our updated manuscript meets your standards. Reviewer 1 had some comments regarding figure 1, so it has been adapted accordingly. There were no other changes to the remaining parts of the manuscript.